

# Autistic traits are associated with individual differences in finger tapping: an online study

Alycia Messing[1] and Deborah Apthorp[1,2]

[1] School of Psychology, University of New England, Armidale, New South Wales, Australia
[2] School of Computing, Australian National University, Canberra, New South Wales, Australia

## ABSTRACT

In a novel online study, we explored whether finger tapping differences are evident in people with autistic traits in the general population. We hypothesised that those with higher autistic traits would show more impairment in finger tapping, and that age would moderate tapping output. The study included a non-diagnosed population of 159 participants aged 18–78 who completed an online measure of autistic traits (the AQ-10) and a measure of finger tapping (the FTT). Results showed those with higher AQ-10 scores recorded lower tapping scores in both hands. Moderation analysis showed younger participants with more autistic traits recorded lower tapping scores for the dominant hand. This suggests motor differences seen in autism studies are evident in the general population.

## INTRODUCTION

Autism is a pervasive developmental condition characterised by persistent differences in communication and interaction across a variety of contexts, as well as repetitive patterns of behaviour and restricted interests (*American Psychiatric Association (APA), 2013*). Prevalence rates in Australia indicate one in 150 are diagnosed with autism (*Australian Institute of Health & Welfare (AIHW), 2017*) with males four times as likely to be diagnosed, consistent with international reporting (*Australian Bureau of Statistics, 2015*, as cited in *Australian Institute of Health & Welfare (AIHW), 2017*; *American Psychiatric Association (APA), 2013*; *Charman, 2002*). Diagnoses of autism have increased considerably in recent years; further, 83% of those diagnosed were young children or young people under 25 (*Australian Bureau of Statistics, 2015*, as cited in *Australian Institute of Health & Welfare (AIHW), 2017*). Many with autism find it challenging to interact with others and have issues with relationships, education and employment (*Australian Institute of Health & Welfare (AIHW), 2017*; *Ashwood et al., 2016*). This has an emotional and financial impact on individuals, families, and communities (*Kogan et al., 2008*). Although social and cognitive deficiencies are core to the diagnosis of autism, an additional element that has become a recent focus is motor skills.

A meta-analysis examining 51 articles which compared autistic individuals to typically developing cohorts showed a significant difference in motor performance (*Fournier et al.,*

Corresponding author
Deborah Apthorp,
dapthorp@une.edu.au

2010). There is also evidence to suggest that the level of impairment in autism correlates with the level of impairment in motor functioning (*Dziuk et al., 2007*). Reports which have seen the most consistent results in motor differences are those using neuropsychological measures of motor impairment (*Duffield et al., 2013*). These motor tasks represent the behavioural output of known motor functions, and can be mapped to regions of the brain, which can provide vital information of neurological and structural underpinnings of autism (*Duffield et al., 2013*; *Mostofsky et al., 2009*). These motor deficits are shaped by early development and then by motor experience over time (*McPhillips et al., 2014*). Additionally, motor studies are not constricted by language or cognitive demands, providing the potential for more representative participant pools (*Mosconi & Sweeney, 2015*). Finally, motor skills develop earlier than social and communication skills, and dysfunction in motor circuits may impact function in parallel circuits responsible for social interaction and communication which are core to autism (*Dziuk et al., 2007*; *Mostofsky et al., 2009*).

Differences have been found in not only gross motor skills but also in fine motor skills in autistic populations (*Fournier et al., 2010*; *Morrison et al., 2018*). Fine motor skills relate to writing, holding small items, buttoning clothing, turning pages, eating, cutting with scissors, and using computer keyboards (*Mosconi & Sweeney, 2015*). One such validated neuropsychological test used to measure fine motor skills is the Finger Tapping Task (FTT; *Gowen & Hamilton, 2013*; *Mosconi & Sweeney, 2015*). This test is thought to be an index of psychomotor speed, which is reflected in everyday tasks such as writing, typing, cooking and speech. Some studies using the FTT have shown impairments in motor functioning in autistic groups when compared to typically developing controls, while others have reported mixed findings or no differences at all (*Behere et al., 2012*; *Duffield et al., 2013*; *Morrison et al., 2018*).

When assessing motor functioning in FTT, differences in age and speed have been reported (*Abu-Dahab et al., 2013*; *Duffield et al., 2013*; *Jansiewicz et al., 2006*). One study which broke down age into young (5–7 years), middle (8–11 years) and older (12–21 years) found that differences in speed increased with age (*Abu-Dahab et al., 2013*). Other studies suggest that although motor impairment becomes more pronounced during maturation from childhood to adolescence, it becomes less identifiable as individuals progress though adulthood (*Fournier et al., 2010*). In contrast, an earlier study of FTT found no significant differences between autistic individuals and typically developing controls aged between 12 and 40 (*Minshew, Goldstein & Siegel, 1997*). More recent research conducted with similar age ranges (5 to 33 years) did find significant speed difference in finger tapping, with autistic individuals tapping more slowly (*Duffield et al., 2013*). Differences in the results could be attributed to difference in methodology, and/or differences in technology or test used and little to no research has been conducted in aging populations (*Duffield et al., 2013*; *Minshew, Goldstein & Siegel, 1997*).

Assessment of dominant and non-dominant hands are a common inclusion in FTT (*Duffield et al., 2013*; *Hardan et al., 2003*). When examining a group of 89 males (59 autistic, 30 controls), *Duffield et al. (2013)* found significant differences in times for non-dominant hand as well as dominant hand finger tapping, $p < 0.01$ and $p < 0.05$

respectively. They also noted that, although their results were significant, they only managed to achieve small effect sizes (*Duffield et al., 2013*). *Hardan et al. (2003)* also found that their control group was significantly faster at the FTT than the autistic group, but only for the non-dominant hand. Other studies assessing finger tapping in dominant and non-dominant hands with a smaller paired male sample failed to find any difference between groups or hands (*Freitag et al., 2007*).

One area with very little research regarding motor differences and autism is the extent to which these impairments are identifiable in individuals from the general population. Autism can be conceptualised as sitting at one end of a continuum with typical development at the other, and thus, autistic traits can be present in individuals in the general population (*Baron-Cohen et al., 2001*; *Ruzich et al., 2015*). Only one study has been published to date; *Cassidy et al. (2016)* explored the relationship between dyspraxia and autistic traits measured by the AQ in a total sample of 8,002 adults (1,237 autistic individuals and 6,765 controls) aged 18–75. Their results indicated that dyspraxia was larger in the autistic group than controls, and that dyspraxia was associated with autistic traits in the control sample (*Cassidy et al., 2016*). However, the study relied on a self-report measure of dyspraxia, which may impact the accuracy due to lack of self-awareness.

A common limitation of these studies is the number of participants; sample sizes for most of the studies have been small, with only more recent research attaining sizes above 50 for each group. Taking research online can be one way to expand and broaden sample sizes while still being as valid and reliable as in person research (*Anwyl-Irvine et al., 2021*; *Bridges et al., 2020*). Another limitation is the age ranges of participants, most focus on children and young adult populations. Very few studies included participants over the age of 40. Thus, there is a gap in understanding of motor impartment in autism in older populations.

Another element not consistently examined is the sex of the participants, with some studies failing to report this completely (*Abu-Dahab et al., 2013*; *Hardan et al., 2003*; *Jansiewicz et al., 2006*). The majority of motor impairment research focuses on males; given the prominence of males diagnosed with autism, this is not unexpected. Recent reviews, however, suggest that sex differences in diagnosis may not be as great as previously thought, as females are better at masking symptoms through compensatory behaviour (*Halladay et al., 2015*; *Livingston & Happe, 2017*). Masking behaviours are unlikely to affect FTT performance; thus, this research could add to our understanding of sex differences in autistic populations by providing a more accurate representation of number of autistic traits in females.

To summarise, finger tapping differences have been found in autistic samples, although most reports have relied on small participant numbers and young male populations. Assessing autistic traits in the general population allows for larger sample sizes, stronger statistical power, and the potential for more even gender representations (*Ruzich et al., 2015*). To date there has only been one study published that explores the relationship between motor impairment and autistic traits in the general population (*Cassidy et al., 2016*). Is it possible to find finger tapping differences in the general population among those who report having autistic traits? This study aimed to answer these questions using a

novel online approach with participants from the general population across a broad range of ages.

In concordance with previous research, we hypothesised that those who reported more autistic traits, scoring higher on the AQ, would show more impairment in motor functioning, having lower FTT scores. We also hypothesised that age would have a moderating effect on FFT scores, such that those who were younger and had higher AQ scores would have lower FTT scores, but that this relationship would be weaker for older individuals. Finally, we aimed to add to our understanding of possible sex differences, and thus we also explored any moderating effect sex might have on the relationship between finger tapping scores and AQ scores.

This study was preregistered at AsPredicted.org: https://aspredicted.org/yi84x.pdf.

# MATERIALS AND METHODS

## Power analysis

An *a priori* power analysis was conducted using G*Power 3.1.9.2 (*Faul et al., 2009*). With a medium effect size (0.15; *Cohen, 1992*), an $\alpha$ of 0.05, and a power of 0.95, it was estimated that a minimum of 129 participants would be required for a multiple linear regression analysis.

## Participants

Following approval by the University of New England's (UNE) Human Research Ethics Committee (Approval no. HE18-164) a total of 331 individuals were recruited as part of a larger group project which focused on sleep, memory, attention and executive function. Only 159 of these completed the full set of tasks required for this experiment. Participants were aged between 18 and 78 ($M = 35.42$, $SD = 14.54$). 112 (70.4%) were female and 47 were male. Participants were a convenience sample recruited online through social media (Facebook, Reddit, and LinkedIn), UNE first-year students, and personal and professional contacts *via* email. Participation in this study was voluntary and no compensation was offered. Participants were required to be aged from 18 to 85, speak English or able to read English, have access to a desktop or laptop computer that had a keyboard and working Internet connection, and be able to provide informed consent.

## Materials

Demographic data was collected based on self-report questions about age, gender, and diagnosed illnesses (see Supplemental Materials).

### AQ-10.

The AQ-10 (*Allison, Auyeung & Baron-Cohen, 2012*) is a 10-item self-report measure used to screen for autistic traits in the general population. The AQ-10 is an adaption of the full 50 item Autism Spectrum Quotient (AQ; *Baron-Cohen et al., 2001*) which uses the two most discriminable items from each of the five subscales. Participants rate the extent to which they agree or disagreed with a statement, such as, "I find it easy to 'read between the lines' when someone is talking to me", by selecting one of four response options "Strongly

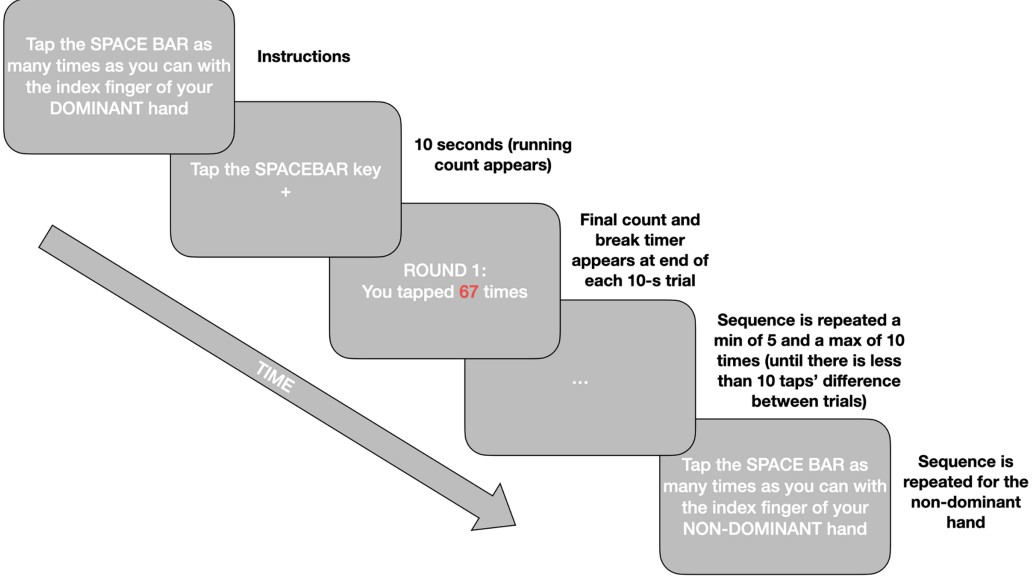

**Figure 1  A simplified schematic of the procedure for the online Finger Tapping Task.**

Agree," "Definitely Agree," "Slightly Disagree," and "Definitely Disagree." A score of one is given for "Strongly Agree," "Definitely Agree" responses. Items 2, 3, 4, 5, 6, and 9 are negatively coded and score of one is given for "Slightly Disagree," and "Definitely Disagree" responses. Total scores can range from 0 to 10. Scores of six or more are indicators that the individual be above the clinical cutoff for autism (*Allison, Auyeung & Baron-Cohen, 2012*). The AQ-10 has been reported as having good reliability and validity with $\alpha = 0.85$ reported in the original validation study (*Allison, Auyeung & Baron-Cohen, 2012*). For the present study, we found considerably lower reliability, with $\alpha = 0.57$. Previous studies have found AQ-10 able to discriminate between clinical and non-clinical samples of autistic individuals (*Allison, Auyeung & Baron-Cohen, 2012*; *Booth et al., 2013*). The AQ-10 sum score has also been shown to be free of sex biases (*Murray et al., 2016*). The AQ-10 was chosen for this experiment for its brevity.

## Finger tapping task

The Finger Tapping Test (FTT; *Shimoyama, Ninchoji & Uemura, 1990*) measures motor function *via* tapping frequency. Participants are required to tap the space bar as many times as possible for 10 s with their index finger. Timing starts with the first tap and only taps recorded during the 10 s are valid. Practice rounds are provided for both hands. The experiment is repeated minimum 5, maximum 10 times for both dominant and non-dominant hands until the scores are within 10 taps of each other. A mandatory 10-s rest is given between each tapping, and 20 s rest after the third tapping. Consistent with the literature final tapping scores, per hand, are the mean of the five rounds which are within 10 taps of each other (*Ashendorf, Horwitz & Gavett, 2015*). The Inquisit script for the online implementation of this test can be found in our Open Materials at https://osf.io/mqyg9/. The procedure is illustrated in Fig. 1.

## Procedure

This study's data was collected as part of a larger project and the process as a whole is reported here for the sake of replicability. This study consisted of two parts—first an online battery *via* Qualtrics (Version August, 2018; *Qualtrics, 2018*) including demographic questions and seven questionnaires, of which the AQ-10 was first (see Supplemental Materials for details). This was followed by a collection of four online experiments presented using Inquisit software (Version 5.0.12.0; *Millisecond, 2018*). Participants accessed the experiment through a link provided on recruitment posts or emails. After reading the Information for Participants section (see Supplemental Materials), they gave implied consent by choosing the "Yes" button below the question "Do you wish to continue?". They then completed the demographic data and questionnaire section, including the AQ-10, Karolinska Sleepiness Scale (KSS; *Akerstedt & Gillberg, 1990*), alcohol use questions from the Alcohol Use Disorders Idnetification Test (AUDIT; *Saunders et al., 1993*), Pittsburgh Sleep Quality Inventory (PSQI; *Buysse et al., 1989*), Morningness-Eveningness Questionnaire (MEQ; *Horne & Östberg, 1976*) and the short-form Depression, Anxiety and Stress Scale (DASS21; *Henry & Crawford, 2005*). Following this, participants were directed *via* hyperlink to another webpage, where they downloaded a program (Inquisit Player) to run the experiments on their own device. Participants were randomly allocated to a running order, (1) FTT, N-back, Psychomotor Vigilance Test and the Number Letter Task, or (2) Psychomotor Vigilance Test, the Number Letter Task, FTT, and N-back. Participants pressed the spacebar to start and proceed between each test. There were no breaks between tests (see Supplemental Materials for screenshots and full instructions of all tests). Participant ID codes were generated in Qualtrics and brought across to Inquisit with the use of a custom URL.

## Data analysis

The data were analysed using the statistical package R, including the following packages: R (Version 4.0.3; *R Core Team, 2020*) and the R-packages *corx* (Version 1.0.6.1; *Conigrave, 2020*), *dplyr* (Version 1.0.7; *Wickham et al., 2021*), ggplot2 (Version 3.3.5; *Wickham, 2016*), *ggridges* (Version 0.5.2; *Wilke, 2020*), *here* (Version 0.1; *Müller, 2017*), *interactions* (Version 1.1.5; *Long, 2019*), lme4 (Version 1.1.27.1; *Bates et al., 2015*), *matrix* (*Bates & Maechler, 2019*), *mosaic* (*Pruim, Kaplan & Horton, 2017*), *papaja* (Version 0.1.0.9997; *Aust & Barth, 2020*), *psych* (Version 2.0.9; *Revelle, 2020*), *raincloudplots* (Version 0.2.0; *Allen et al., 2021*), *readr* (Version 2.0.0; *Wickham & Hester, 2020*), *tableone* (Version 0.13.0; *Yoshida & Bartel, 2020*), and *tidyverse* (Version 1.3.0; *Wickham et al., 2019*). Custom R code was used to link the data from the two platforms (Qualtrics and Inquisit)—see the OSF repository at https://osf.io/mqyg9/ for a copy of this code. Due to attrition before the Inquisit experiment, an independent samples t-test was run to compare those who partially and fully completed the experiment assessing key experimental variables of AQ-10 score, age and gender (coded one for female, two for male). The Shapiro-Wilk test of normality indicated the samples were not normally distributed, AQ-10 score ($W = 0.94$, $p < 0.001$), age ($W = 0.91$, $p < 0.001$), gender ($W = 0.61$, $p < 0.001$), and therefore the

**Table 1 Descriptive statistics of the key variables overall and stratified by gender.**

|  | Overall | Female | Male | $p$ |
|---|---|---|---|---|
| $n$ | 158 | 111 | 47 | |
| Age (years) (mean ($SD$)) | 35.41 (14.54) | 35.01 (14.48) | 36.36 (14.78) | 0.594 |
| AQ-10 score (mean ($SD$)) | 3.08 (1.82) | 3.02 (1.91) | 3.23 (1.59) | 0.496 |
| Dominant hand score (mean ($SD$)) | 66.38 (9.75) | 64.46 (8.50) | 70.91 (11.03) | <0.001 |
| Non-dominant score (mean ($SD$)) | 61.07 (8.32) | 59.81 (8.30) | 64.06 (7.66) | 0.003 |

non-parametric Welch's test was used. There were no significant differences between groups for any of the variables.

Only the fully completed data was included for further analysis. There were no missing values and only one participant's (36-year-old F) responses on the FTT fell out of acceptable range, 2.0 for dominant and 2.2 for non-dominant, and was excluded from further analyses. All other responses were included as there was no evidence the responses were not valid. Additionally, participants who indicated they had been diagnosed with a neurological condition (two female) or ADHD/ADD (three male) were not excluded as doing so did not change any significant results.

## RESULTS

### Descriptive statistics

Mean and standard deviation scores for the key variables are displayed in Table 1.

### Preliminary analysis

A paired samples t-test was conducted to compare hand tapping in dominant and non-dominant hands. Shapiro-Wilk test of normality indicated the sample was not normally distributed ($W = 0.98$, $p = 0.019$) therefore the non-parametric Wilcoxon W test was used. There was a significant difference between dominant ($M = 66.38$, $SD = 9.75$) and non-dominant ($M = 61.07$, $SD = 9.75$) hand tapping scores, $W = 11,045$, $p < 0.001$, 95% CI = [4.10–6.00], $d = 0.82$ (Fig. 2). Due to this, all statistical tests were conducted separately for each hand.

### Correlational analysis

As assumptions of normality were violated, Spearman's correlation analyses were run for each hand to assess the associations between AQ-10 score, age, gender and tapping scores (see Table 2).

Based on these analyses, age was negatively associated with AQ-10 score and only gender was found to be positively correlated with tapping scores for both hands (such that male participants had higher tapping scores than females—also evident in Table 1). These correlations reflected small to medium effect sizes (*Cohen, 1988*).

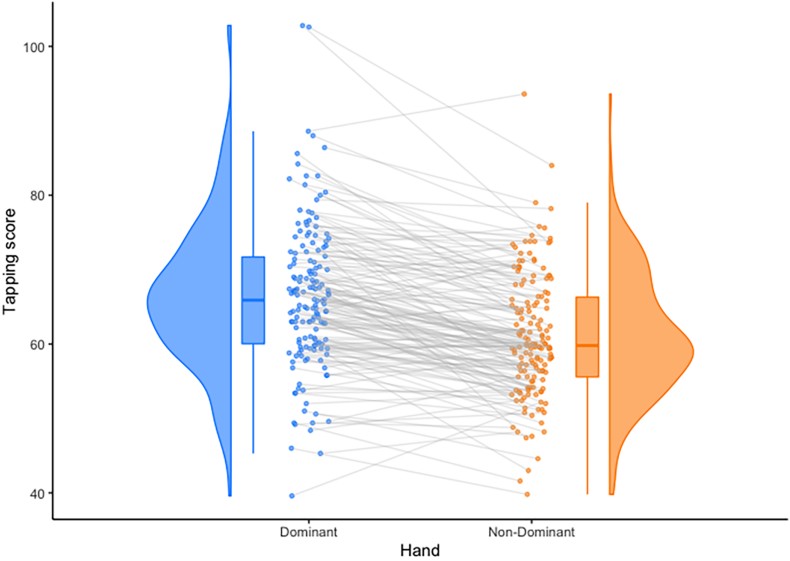

**Figure 2 Observed differences between mean hand tapping scores.** Individuals' scores are connected with grey lines, and distributions are shown as probability density functions on the outside of the plots. Means are depicted by solid lines within the bars, and bars depict upper and lower quartiles. Plotted using the Raincloud Plots toolbox in R (*Allen et al., 2021*).   

**Table 2 Zero-order correlations.**

|  | 1 | 2 | 3 | 4 |
| --- | --- | --- | --- | --- |
| 1. Age | – |  |  |  |
| 2. Gender | 0.06 (0.462) | – |  |  |
| 3. AQ-10 score | −0.36 (<0.001) | 0.09 (0.292) | – |  |
| 4. Tapping score dominant hand | −0.16 (0.048) | 0.27 (<0.001) | −0.07 (0.333) | – |
| 5. Tapping score non-dominant hand | −0.07 (0.370) | 0.24 (0.002) | −0.10 (0.167) | 0.73 (<0.001) |

**Notes:**
Correlations between key variables and tapping scores.
*P*-values are in brackets.

## Regression analysis

Two general linear model analyses (GLMs) were carried out to investigate the influence autistic traits in the general population (AQ-10 scores), age, and gender have on FTT scores. Both Q-Q plots showed that the data was normally distributed, Kolmogorov-Smirnov test and the Shapiro-Wilk's W test were both not significant ($W = 0.984$, $p = 0.059$, $D = 0.054$, $p = 0.743$ dominant; $W = 0.948$, $p = 0.057$, $D = 0.083$, $p = 0.226$ non-dominant). Levene's test indicated no violations to the assumption of homogeneity of variance for either hand ($p = 0.114$ dominant, $p = 0.235$ non-dominant). Following the recommendations of *Cohen et al. (2013)*, AQ-10 scores and age were centred at zero and gender was dummy-coded.

The overall models were both statistically significant, explaining 22% of the variance for dominant hand, $R^2 = 0.22$, 90% CI [0.12–0.31], $F(4,153) = 11.04$, $p < 0.001$, and 12%. of the variance for the non-dominant hand, $R^2 = 0.12$, 90% CI [0.03–0.19], $F(4,153) = 5.20$,

**Table 3 Regression results for the dominant hand.**

| Predictor | $b$ | 95% CI | $t(153)$ | $p$ |
|---|---|---|---|---|
| Intercept | 82.63 | [74.96–90.30] | 21.28 | <0.001 |
| AQ 10 score | −3.79 | [−5.78 to −1.79] | −3.74 | <0.001 |
| Age | −0.39 | [−0.58 to −0.20] | −4.08 | <0.001 |
| Gender | 6.44 | [3.41–9.48] | 4.19 | <0.001 |
| AQ 10 score × Age | 0.07 | [0.01–0.13] | 2.48 | 0.014 |

Notes:
Model regressing gender, AQ-10 score, and age on dominant hand tapping scores.
$R^2 = 0.22$, 90% CI [0.12–0.31], $R^2_{adj} = 0.20$.
B, unstandardised regression coefficient; SE, standard error; β, standardised regression coefficient; CI, confidence interval; t, t-test value; p, probability.

**Table 4 Regression results for the non-dominant hand.**

| Predictor | $b$ | 95% CI | $t(153)$ | $p$ |
|---|---|---|---|---|
| Intercept | 70.22 | [63.24–77.20] | 19.88 | <0.001 |
| AQ 10 score | −2.13 | [−3.95 to −0.31] | −2.32 | 0.022 |
| Age | −0.21 | [−0.38 to −0.04] | −2.44 | 0.016 |
| Gender | 4.32 | [1.55–7.08] | 3.09 | 0.002 |
| AQ 10 score × age | 0.04 | [−0.02 to 0.09] | 1.37 | 0.173 |

Notes:
Model regressing gender, AQ-10 score, and age on non-dominant hand tapping scores.
$R^2 = 0.22$, 90% CI [0.12–0.31], $R^2_{adj} = 0.20$.
B, unstandardised regression coefficient; SE, standard error; β, standardised regression coefficient; CI, confidence interval; t, t-test value; p, probability.

$p = 0.001$. Gender, AQ-10 score, age, and the interaction between AQ-10 and age explained significant amounts of unique variance in finger tapping scores in the dominant hand (see Table 3). Only gender, AQ-10 score, and age explained significant amounts of unique variance in finger tapping scores in the non-dominant hand (see Table 4). When age or AQ-10 scores increased, tapping decreased for both hands.

To interpret the interaction in the dominant hand, a simple slopes analysis was carried out, illustrated in Fig. 3. Statistical test values for this analysis are shown in Table 5. Examination of the simple slope coefficients revealed that for the younger participants (−1SD), there was a significant negative linear relationship between AQ-10 scores and tapping scores. When participants were closer to the mean age there was also a significant negative linear relationship between AQ-10 scores and tapping scores, but with a shallower slope. For older participants (+1SD), there was no significant relationship between AQ-10 scores and tapping scores. This suggest that as age increases, the relationship between tapping score and AQ-10 score weakens.

## DISCUSSION

The purpose of the present study was to explore the possibility that differences in fine motor skills would be evident in individuals of the general population who exhibit autistic traits, as measured by the AQ-10 and FTT, as well as to look at any impact age and gender might have on this relationship. The results partially supported our predictions.

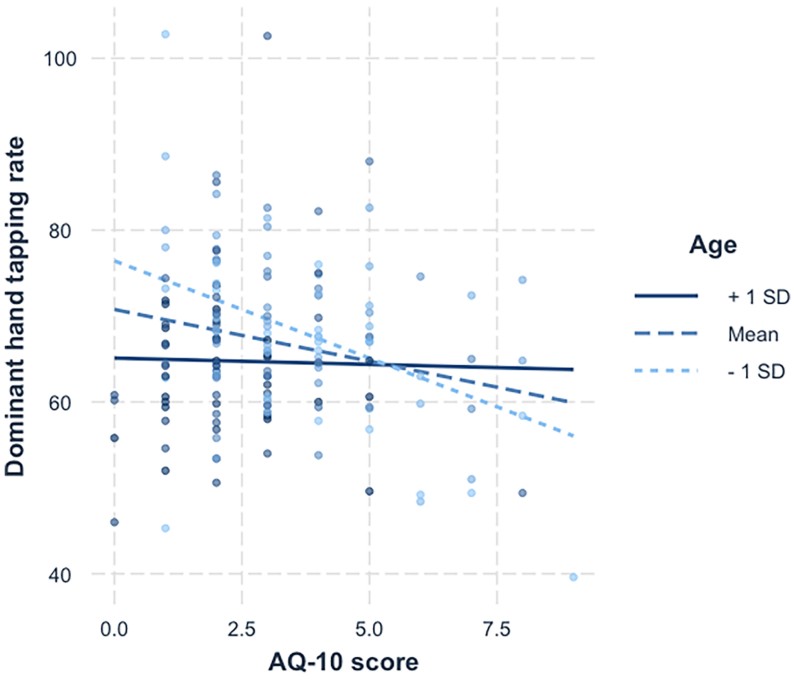

**Figure 3 Simple slopes for age x AQ-10 score.** A simple slopes plot illustrating the interaction between age and AQ-10 score for the dominant hand tapping data.

**Table 5 Simple slopes analysis.**

| Value of age | Est. | S.E. | 2.5% | 97.5% | t val. | p |
|---|---|---|---|---|---|---|
| 20.87 | −2.26 | 0.52 | −3.28 | −1.25 | −4.39 | <0.001 |
| 35.41 | −1.21 | 0.42 | −2.04 | −0.37 | −2.84 | 0.01 |
| 49.95 | −0.15 | 0.68 | −1.48 | 1.19 | −0.22 | 0.83 |

Note:
A simple slopes analysis of the moderating effect of age on the relationship between AQ-10 scores and tapping scores, using Johnson-Neyman intervals.

The results showed that there was a significant relationship between finger tapping scores and autistic traits, with those who had higher autistic traits showing lower tapping scores. This was found in both hands, although the effect was larger in the dominant hand. The second hypothesis, that those who were younger and had more autistic traits would have a lower FTT score, was partially supported, as a significant interaction was found only in the dominant hand results. Our final hypothesis exploring the moderating effect sex has on the relationship between finger tapping scores and AQ scores was not significant for either of the GLMs, suggesting there is no evidence for an interaction between sex and autistic traits when predicting finger tapping scores in the general population.

The results of the current study reveal that impairment in fine motor skills can be seen in members of the general population who exhibit autistic traits. This aligns with previous research which found that individuals with autism perform worse than controls on measures of fine motor skills (*Duffield et al., 2013*; *Fournier et al., 2010*; *Morrison et al., 2018*). Like other core diagnostic domains, motor differences can be identified in

non-clinical populations, providing further support for the conceptualising of autism as a continuum (*Baron-Cohen et al., 2001*). The presentation of the significant results in each hand is reflective of results found in *Duffield et al. (2013)*, where the amount of variance explained was greater for dominant hand than non-dominant. As there have been limited studies focusing on motor functioning in adulthood, and the current findings are the first to explore motor impairment and autistic traits in the general population, they should be interpreted with caution. However, they do add to the growing body of research showing neuromotor differences in conjunction with autistic traits in adult populations.

One possible explanation for these findings is the observed link between autism and specific neural abnormalities and the processes these areas are believed to be involved in. Converging evidence based on neurological studies indicates deficits in the cerebellum, the basal ganglia, the medial temporal lobe, and the frontal lobe in autism (*Penn, 2006*; *Tsai, 2005*). FTT performance has been associated with activation in the primary sensorimotor cortices, supplementary motor area, premotor cortex, inferior parietal cortices, basal ganglia, and cerebellum (*Witt, Laird & Meyerand, 2008*). It is possible that damage or changes to these areas may be responsible for the observed association of impaired motor performance and autistic traits (*Nayate, Bradshaw & Rinehart, 2005*).

The moderating effect of age on AQ-10 scores and finger tapping scores suggests that motor impairment is more evident in younger individuals who have more autistic traits. This was only found in the dominant hand, becoming less pronounced in older participants, and is not significant in participants 1 SD above the mean age (see Fig. 3). It should be noted that the mean age for the 'younger' group was almost 21, and for 'older' was close to 50. These results are consistent with previous studies showing that although motor impairment in individuals with autism becomes more pronounced during maturation from childhood to adolescence, it becomes less identifiable as individuals progress though adulthood (*Fournier et al., 2010*). Explanations for this difference in speed are linked to neuroplasticity, suggesting alternate pathways are being utilised, resulting in a potential delay in processing (*Duffield et al., 2013*).

When compared to finger-tapping age trends in the general population, this study is consistent with the literature in that, as people age, tapping slows (*Hubel et al., 2013*). It is possible that age-related changes are stronger in older populations and obscure any potential effect autistic traits have on motor skills. One theory proposed by *Lever & Geurts (2016)* suggests that autistic individuals cope better with ageing, or are protected against some cognitive ageing effects. In their study, individuals with autism scored higher than neurotypicals on visual memory, particularly in the older (50+) age group, and theory of mind differences in the younger group were absent in the older group. However, this model has not been tested in the domain of motor cognition. In this light, the present results could be interpreted as showing that those who do not display autistic traits are slowing down more, to a similar speed as those with autistic traits. However, further research into ageing and older populations is needed before any conclusions can be drawn. Additionally, the negative correlation age has with the AQ-10 indicates there may be changes in autistic traits as people age. These results may be due to age-related changes, or to failure of the AQ-10 to accurately capture autism in that population. Contrary to these

results, one study found that AQ scores of autistic individuals increase with age (*Happé et al., 2016*). Again, further research into ageing populations is warranted prior to any definitive conclusions being drawn.

## Limitations and future directions

Although the current study aimed to explore motor impairment and autistic traits in the general population, it did not contain any participants who identified as autistic. Having such participants would have potentially strengthened these results, if the same pattern was also found in that sample, and by adding to the number of higher scoring individuals. However, this does not invalidate the interesting results in a non-clinical population, as autistic traits are thought to be on a continuum throughout the population. Future research should aim to recruit from the general populations and autism-diagnosed populations.

Additionally, the age range of this sample cannot fully address the substantial gap in the literature regarding the manifestation of motor impairment in individuals with high autistic traits in the ageing population. Although the oldest participant in the study was in 78, only 31 participants (19.6%) were over the age of 50. This, taken in conjunction with the non-significant results in this group in the simple slopes analysis (possibly due to low power), limits the generalisability of these results to older populations. Given the changes that occur as people age, and the potential that these are different in autistic samples, longitudinal studies would be more informative. Such research would greatly add to our understanding of ageing in autistic populations.

Attrition was a major issue in this project. Although the analysis showed no significant differences on key demographic variables, this does not mean the participants did not differ in other ways which may have impact on the results. It is suspected the requirement of downloading a platform proved to be a deterrent to many. Future online research of this kind should aim to use a single link or platform to complete the study which may improve completion rates. Furthermore, the use of online research like this we ran the risk of individuals not following the instructions fully: for instance, someone not using the correct finger, or using more than one finger; and there was no way for us to police this practice. Although we only had one outlier in this study, we acknowledge we acknowledge that the data were not collected under controlled conditions. However, this would likely have added noise to the data rather than biasing it systematically.

We did not measure handedness in this study, instead relying on participants to use their dominant and non-dominant hands in the respective conditions. In future, a brief measure of handedness such as the Flinders Handedness Survey (FLANDERS; *Nicholls et al., 2013*) might be useful to add.

This was the first study to look at motor impairment with the FFT and autistic traits in the general population. The FFT is only one form of testing motor functioning, although it is considered to be a purer test, without visuospatial input (*Hubel et al., 2013*).
The replication of this study with other measures of fine motor functions would be useful in providing support to the existence of motor impairment in not only autistic populations but people of the general population who display autistic traits. We also recognize that the

AQ-10, although a recommended tool, is not without its critique (see *Taylor et al. (2020)*) and for this study the α was low. In addition, some studies have suggested that online use of the AQ-10, compared to face-to-face use, can affect reliability and validity (*Cheung et al., 2023*). This survey was chosen for its brevity; future research may wish to look to other measures of autistic traits, such as the CATI (*English et al., 2021*).

## CONCLUSIONS

This study set out to explore whether finger tapping differences, found in autistic populations, are identifiable in individuals from the general population who exhibited autistic traits. The results suggest that not only do they exist, but there is an effect of age, in that these tapping differences becomes less identifiable as individuals progress though adulthood. As this was one of the first of such studies looking at autism and motor impairment in the general population, and using an online platform to do so, replication with other fine more skill tests, as well as further exploration of the affects ageing has on motor functioning and autistic traits, is essential to fully understand this relationship. Better understanding of the functioning of fine motor skills in the autistic and general populations could not only assist in early diagnosis of autism, but also improve our understanding of ageing, and in turn, help direct treatment and support. In terms of practical applications, this is a quick and easy-to-obtain measure that might be useful as a screening tool for clinicians, in conjunction with other tests.

### Open data and code

The data and code to reproduce the results in this article are all available publicly at https://github.com/deborah-apthorp/FTT_Autism_Reproducible_Results. Experiment code and R code to reproduce the full manuscript are available at https://osf.io/mqyg9/.

## ACKNOWLEDGEMENTS

We would like to thank Dr. Amanda Bolbecker for her helpful comments on the study.

### Funding

The authors received no funding for this work.

### Competing Interests

The authors declare that they have no competing interests.

### Author Contributions

- Alycia Messing conceived and designed the experiments, performed the experiments, analyzed the data, authored or reviewed drafts of the article, and approved the final draft.
- Deborah Apthorp conceived and designed the experiments, analyzed the data, prepared figures and/or tables, authored or reviewed drafts of the article, and approved the final draft.

## Human Ethics

The following information was supplied relating to ethical approvals (*i.e.*, approving body and any reference numbers):

Ethical approval was granted by the University of New England's Human Ethics Research Committee.

## Data Availability

The data and code are available at GitHub and OSF: https://github.com/deborah-apthorp/FTT_Autism_Reproducible_Results Messing, Alycia, and Deborah Apthorp. 2023. "Motor Differences in Individuals with Autistic Traits–Data and Code." OSF. April 26. DOI 10.17605/OSF.IO/MQYG9.

## Supplemental Information

Supplemental information for this article can be found online at http://dx.doi.org/10.7717/peerj.15406#supplemental-information.

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
