# Peer review of "Autistic traits are associated with individual differences in finger tapping: an online study"

_PeerJ, doi:10.7717/peerj.15406_

## Round 0.1 · original submission · Major Revisions

Dear authors,

please reply point by point to the reviewers' comments.

·

Basic reporting

Dear Authors, I appreciate your work on this interesting study. In this review I would like to help You increase the quality of the work, which I already consider very high.

Formatting – it looks much better and is easier to read when the text is adjusted.

In my view the title “Motor differences in individuals with autistic traits” is way to vast, because there is one single test used in the study – the finger tapping test. This test is specific to assess the speed of action of fingers in dominant and non dominant hands only. There are no tests used to assess other motor skills like strength, stamina, coordination, dexterity, mobility etc. Secondly – it is needed to state in the introduction that the conditions were online, and finally it may be considered to add the information that the participants were not diagnosed with having autistic disorders.
Suggested title to consider:
Differences in finger tapping tests results are correlated with the autistic disorders scale and age in non-diagnosed populations with autistic traits. An online, self assessment study.

English is professional trought the whole paper.
Literature refferences are uptodate and sufficient.

the structure of the article is professional, the quality of the tables and figures is high.

There is no raw data shared that can be downloaded at the review stage.

Experimental design

The aims and hypothesis are well formulated, but not fulfill the title of the paper, so I proposed the change of the title.
Methodology section need more details and information to replicate and clearly show conditions and limitations.
I strongly recommend to include in the title and aims the element of online testing and online self assessment. I understand that the FTT results did not fell out of acceptable range, even though it is important to charactierize the limitations: The conditions of conducting the test are unknown and it can affect the results. It is clear that researchers understand these limitations, but it needs to be underlined in the text so that the readers fully understand that the conditions were not controlled due to online conditions. F.e. the technique of the FTT – one can do the test with the wrist on the laptop case, some with the wrist on the table using separate keyboard, some with the wrist in the air. What is more the type and construction of the keyboard could possibly affect the results, one possibly could use the same hand for dominant and non-dominant test, and finally – the situation where somebody uses two hands or more fingers to do the test neither can be declined.
If the authors find a paper comparing the online AQ-10 results with face to face self assessment or assessment by a third person it may be considered to cite in the work to show the possible differences coming from the different conditions of the assessment.
I recommend putting information about AQ-10 questtions and characterize it clearly to the readers in the methods or introduction section.

Data analysis
I suggest considering putting the data about the software used to an appendix at the end of the paper.

the ethical standards are proven.

Validity of the findings

Conclusions
I reccoment shortening the conclusions section and adding a section “practical application”. There is no link between the FTT and every day life fine motor skills and actions characterized in the introduction. If the Authors would do so, at the end of the paper in the practical application section there can be put very usefull instructions for the practicioners.

Additional comments

The quality of the paper is high, but I pointed out some aspects which can be improved. The practical aplication section is missing and I strongly recomend to add it to the paper.

Reviewer 2 ·

Basic reporting

- The introduction, literature, and relevance of the study are written and justified clearly. The authors discuss the need to study motor functioning in general populations with ASD traits. They also discuss age and sex as moderators that may affect motor functions.
- Line 109: authors stated that “To date there has only been one study published that explores the relationship between motor impairment and autistic traits in the general population”. However, no citation is provided.
- The hypothesis is clearly written.
- All statistical notations should be in italics. For example, in Table 1 (p, SD)
- Line 307: The idea proposed by Lever and Geurts (2016) sounds interesting. It is better if the author briefly elaborates on the theories.
- Line 347 and 348: Can specifically mention other reliable measures of autistic traits.
- Line 352: it should be the general population.

Experimental design

- For the Participants section, it is unnecessary to inform the reader that “a total of 331 individuals were recruited as part of a larger group project which focused on sleep, memory, attention and executive function.” I suggest just stating how many participants are recruited for this study, which is 159.

Validity of the findings

- The author did not mention how they measure the dominant vs non-dominant handedness. What instrument are they using?
- It would be good if the author could report how many participants are under High ASD traits and Low ASD traits. We can see the pattern of participants’ AQ-10 scores in Figure 2, but the exact number of High and Low ASD traits is not explicitly stated.
- The Analysis and Results section did not answer and is not parallel to the research hypothesis. In the hypothesis, the author mentioned that they predict and want to compare high and Low ASD traits on motor functioning. And also the effect of age and gender. It would be good if the author could add the comparison analysis of Low and High ASD traits on motor functions.
- A descriptive table showing the age distribution of the participants can be added since age is an important variable in this study. Is the number of young and old participants appropriate to make a comparison?

Additional comments

- In this study, the motor function measures are limited to Finger Tapping Task (FTT). It is insufficient. Could you justify why you rely on this one measure and elaborate further on why you chose (FTT)?
- The novelty of this study is that the author tried to show the difference in motor functions in high and low individuals with ASD traits from the general population. However, this aspect is not sufficiently analysed and discussed in the report.
- Methods section, especially Procedure, should include more details.

Reviewer 3 ·

Basic reporting

The article “Motor differences in individuals with autistic traits” presents a clear methodology for the analysis of a sample of individuals with autism traits (none with ASD diagnosis) using the finger tapping task and AQ10 score. The objectives and hypotheses are clearly stated as well as discussion of related literature.

Experimental design

An important concern with the experimental design is the use of the AQ10 score, which have been reported as having poor internal reliability, particularly when applied on non-ASD diagnosed individuals. Please, cite reference [1] below.

References:
[1] Taylor, Emily C., Lucy A. Livingston, Rachel A. Clutterbuck, Punit Shah, and Christine Payne. "Psychometric concerns with the 10-item Autism-Spectrum Quotient (AQ10) as a measure of trait autism in the general population." Experimental Results 1 (2020).

I imagine, the reliability analysis was conducted with this sample; however, details do not appear in the article.

Validity of the findings

My other main concern is about the methodology because it seems that the variable “Gender” may have a critical role in the final conclusions. This variable is discarded from the main analysis using Johnson-Neyman’s simple slopes and also from the analyses shown in Tables 3 and 4, where its interaction with AQ10 score is not explored as was done for Age. Notice that there is a significant difference in the finger tapping task (FTT) performance for Gender. Thus, it could have a significant influence on the results plotted in Fig. 2 where the relationship between FTT rate and AQ10 score is studied in more detail while only controlling by Age.
In general, instead of Johnson-Neyman interval analysis, I would suggest using “linear regression with interaction effects” analysis, which can explore Gender together with Age and all the possible quadratic interactions. For example, here online it can be seen an application of stepwise linear regression, which may allow a better analysis of this dataset (note that all quadratic interactions among predictors are considered in the model):
https://uk.mathworks.com/help/stats/linear-regression-with-interaction-effects.html
An alternative would be to substitute the analysis in Tables 3 and 4 with the suggested more general “interaction effects” analysis (including all possible quadratic interactions), and in the case that only Age’s interactions are significant, then proceed with the Johnson-Neyman analysis.

Additional comments

The article is well written and references are correctly used. Overall, I think it is an interesting article.

---

## Round 0.2 · Minor Revisions

My name is Nathan Caruana - I have just taken over the handling of your manuscript as the original handling editor is no longer available.

To minimise further delays in the review of your manuscript, I have carefully reviewed your revised manuscript and rebuttal letter.

It is my assessment that you have adequately addressed most of the reviewers' comments and so I am happy to suggest one more round of minor revisions without sending out again for another round of review.

I have attached an annotated version of your rebuttal letter where I highlight the changes that I think are still necessary. I have also added another comment at the very beginning concerning language used to describe autism, which I hope you will consider addressing in your revisions.

I look forward to receiving your revisions. Please include a new rebuttal letter in your submission addressing the outstanding issues I have noted.

Best,
Nathan

---

## Round 0.3 · accepted · Accept

Hello,

I am happy with the revisions that were made. May I request that you note two minor corrections at the proofing stage?

1. suggest using "non-autistic" instead of "neurotypicals"
2. instead of "in conjunction with other *tests* --> i would say "diagnostic assessments" since autism diagnoses are also based on general clinical assessments/observations and not necessarily 'tests' per se.

I will leave this to you at the proofing stage and in the meantime, congratulations.

Nathan Caruana